# The Role of Chloroviruses as Possible Infectious Agents for Human Health: Putative Mechanisms of ATCV-1 Infection and Potential Routes of Transmission

**DOI:** 10.3390/tropicalmed8010040

**Published:** 2023-01-05

**Authors:** Yury V. Zhernov, Sonya O. Vysochanskaya, Artem A. Basov, Vitaly A. Sukhov, Anton A. Simanovsky, Inna A. Fadeeva, Roman V. Polibin, Ekaterina A. Sidorova, Denis V. Shcherbakov, Oleg V. Mitrokhin

**Affiliations:** 1Department of General Hygiene, F. Erismann Institute of Public Health, I.M. Sechenov First Moscow StateMedical University (Sechenov University), 119435 Moscow, Russia; 2Center of Life Sciences, Skolkovo Institute of Science and Technology, 121205 Moscow, Russia; 3Department of Chemistry, Lomonosov Moscow State University, 119991 Moscow, Russia; 4Center for Medical Anthropology, N.N. Miklukho-Maclay Institute of Ethnology and Anthropology of the Russian Academy of Sciences, 119017 Moscow, Russia; 5Department of Medical and Biological Disciplines, Reaviz Medical University, 107564 Moscow, Russia; 6Diphtheria and Pertussis Surveillance Laboratory, G.N. Gabrichevsky Research Institute for Epidemiology and Microbiology, 125212 Moscow, Russia; 7Department of English Language, Institute of World Economy, Diplomatic Academy of the Russian Foreign Ministry, 119034 Moscow, Russia; 8Department of Public Administration in Foreign Policy, Diplomatic Academy of the Russian Foreign Ministry, 119034 Moscow, Russia; 9Department of Epidemiology and Evidence-Based Medicine, F. Erismann Institute of Public Health, I.M. Sechenov First Moscow State Medical University (Sechenov University), 119991 Moscow, Russia

**Keywords:** Chlorovirus, ATCV-1, Chlorella, neuroinflammation, cognitive function, nervous system infections, freshwater sources

## Abstract

The *Chlorovirus* genus of the *Phycodnaviridae* family includes large viruses with a double-stranded DNA genome. Chloroviruses are widely distributed in freshwater bodies around the world and have been isolated from freshwater sources in Europe, Asia, Australia, and North and South America. One representative of chloroviruses is Acanthocystis turfacea chlorella virus 1 (ATCV-1), which is hosted by *Chlorella heliozoae*. A few publications in the last ten years about the potential effects of ATCV-1 on the human brain sparked interest among specialists in the field of human infectious pathology. The goal of our viewpoint was to compile the scant research on the effects of ATCV-1 on the human body, to demonstrate the role of chloroviruses as new possible infectious agents for human health, and to indicate potential routes of virus transmission. We believe that ATCV-1 transmission routes remain unexplored. We also question whether chlorella-based nutritional supplements are dangerous for ATCV-1 infections. Further research will help to identify the routes of infection, the cell types in which ATCV-1 can persist, and the pathological mechanisms of the virus’s effect on the human body.

## 1. Introduction

The *Chlorovirus* genus of the *Phycodnaviridae* family includes large viruses with a double-stranded DNA genome [1]. The natural hosts of these viruses are single-celled green algae of the *Chlorella* genus, with a particular virus species infecting a specific algae species. Chloroviruses are widely distributed in freshwater bodies around the world and have been isolated from freshwater sources in Europe, Asia, Australia, and North and South America [2,3]. One representative of chloroviruses is Acanthocystis turfacea chlorella virus 1 (ATCV-1), hosted by *Chlorella heliozoae*. A few publications in the last ten years about the potential effects of ATCV-1 on the human brain sparked interest among specialists in the field of human infectious pathology.

It is known that in the human body, there are no sterile organs where there would be no microorganisms that belong to different kingdoms of viruses, bacteria, or fungi and together make up the human microbiome. Many viruses can colonize trans-barrier organs. For example, the Zika virus can pass through the blood–placental barrier and lead to neonatal microcephaly [4]. The human immunodeficiency virus (HIV) can cross the blood–brain barrier and infect microglial cells [5], and its ability to do so is dependent on the emergence of new recombinant forms of the virus [6,7]. The discovery of new viruses capable of passing through trans-barrier organs would shift our perspective on them from obligate parasites to a permanent viral community known as the virome. Studies in both animal models and humans have shown that the microbiome (and possibly the virome) affects numerous biological functions, including cognitive function [8,9].

The goal of our viewpoint was to compile the scant research on the effects of ATCV-1 on the human body, to demonstrate the role of chloroviruses as new possible infectious agents for human health, and to indicate potential routes of virus transmission.

## 2. Association of the ATCV-1 with Mental and Psychiatric Disorders

In the metagenomic sequencing of DNA samples obtained from the oropharynx of 33 adults without known psychiatric disorders or diseases, Yolken et al. unexpectedly revealed a significant number of reads homologous to various regions of the genome of the ATCV-1 [10]. Virus sequences were identified in 14 out of 33 subjects (42.4%). This virus is common in the aquatic environment, but prior to this work, ATCV-1 had not been shown to be able to infect humans or animals or colonize human mucous membranes. Further, to confirm the presence of ATCV-1 in the oropharynx of the subjects, the authors used the method of detecting the DNA of the ATCV-1 virus with quantitative PCR (qPCR) using fluorescent probes (Taqman). They developed primers specific for the *z100l* gene of the ATCV-1 virus. qPCR analysis detected ATCV-1 DNA in all 10 subjects whose samples contained two or more reads homologous to the viral genome and in 12 of 14 subjects with single reads homologous to the ATCV-1 genome. The results of the analysis showed negative results with human DNA, the buffer solution used, as well as ATCV-1 host DNA from *Chlorella heliozoae*, other PBCV-1 and CVM-1 chloroviruses, and their host organisms: *Chlorella variabilis* and *Micractinium conductrix* [10]. The passage of negative controls indicates the high specificity of the developed method for the detection of ATCV-1 DNA.

The proposed qPCR method was applied to oropharyngeal samples from a larger group of 92 individuals, including 33 previously tested subjects. In 40 (43.5%) of 92 samples, ATCV-1 DNA was detected. At the same time, the DNA of the ATCV-1 virus was not detected in the blood in any of the tests. There were no statistically significant differences between subjects with and without viral DNA in demographics such as age, gender, ethnicity, education level, smoking, basal metabolic rate (BMR), travel history outside North America, and place of birth. In addition, the authors of the work evaluated the relationship between the detection of ATCV-1 DNA and the success of passing a battery of cognitive tests. A statistically significant association was found between the presence of ATCV-1 DNA and lower scores on the Visual Reaction Speed Test, Trail Making Test Part A (*p* < 0.002), Delayed Memory (*p* < 0.039), and Attention (*p* < 0.011) from the Repeatable Battery for the Assessment of Neuropsychological Status (RBANS) as well as the total scores in this battery of tests (*p* < 0.014).

In addition to the human study, Yolken et al. created a mouse model of ATCV-1 infection. They orally administered *Chlorella heliozoae* alone (to a control group of 20 mice) or *Chlorella heliozoae* infected with ATCV-1 (to an experimental group of 30 mice) to mice. During the experiments, the memory of spatial recognition was tested in a Y-shaped maze. Mice exposed to ATCV-1 were less likely to enter a new, previously closed maze arm (*p* = 0.015), while the levels of locomotor activity did not differ between groups. Testing of recognition memory showed a decrease in the time spent exploring a new object by mice from the experimental group (*p* < 0.001) and a familiar object in a new location (*p* = 0.008). At the same time, the passive avoidance test, which tests memory for negative stimuli, did not show statistical differences. In tests with an open field and a dark–light box on the level of anxiety, there were also no differences between the groups. In addition, mice exposed to ATCV-1 were found to have worse pre-pulse inhibition (*p* < 0.05), which may reflect the deterioration in sensorimotor filtration that is characteristic of many psychiatric and neurological diseases.

Despite fairly convincing evidence, the detection of the ATCV-1 virus was questioned in an article by Kjartansdóttir et al. [11]. This commentary to Yolken et al. points out the possibility of the accidental contamination of laboratory reagents with ATCV-1 DNA. In response, the authors of the original article point to the ongoing negative controls and the use of several different methods in principle and reagents used to detect the virus and the fact that the molecular data for the article were obtained in two different laboratories [12].

## 3. Putative Molecular Mechanisms of ATCV-1 Infection

To study the molecular processes behind the changes in the behavior and memory of mice exposed to the ATCV-1 virus, the authors of the work investigated gene expression in the hippocampus, a region of the limbic system of the brain that plays a key role in learning and memory [10]. Significant differential expression of 1285 individual genes was identified. Among the signaling pathways that have been documented to explain behavioral disturbances are dopamine receptor signaling, cyclin-dependent kinase 5 (Cdk5) signaling, antigen presentation, immune cell adhesion, and eukaryotic initiation factor 2 (eIF2). Thus, it is known that changes in the dopaminergic system can lead to a deterioration in the recognition of new objects and pre-pulse inhibition, that the Cdk5 pathway is one of the key pathways in the formation of memory and the regulation of synaptic plasticity, and that eIF2 controls the protein synthesis process necessary for long-term memory. These facts may explain the deterioration in memory in the mice exposed to ATCV-1.

In addition to the identified changes in the pathways associated with the activity of microglia and other immune cells, specific antibodies against ATCV-1 were detected in the experimental group of mice. These data may suggest that infection with ATCV-1 is followed by a pronounced immune response, which may lead to the activation of microglia and the release of pro-inflammatory cytokines that can affect memory and cause cognitive deficits through neuroinflammation mechanisms. In more recent work on the interaction of ATCV-1 with mammals, the virus was directly injected intracranially. There was a significant increase in neuroinflammation markers such as IL-6, iNOS, IFN-γ, and CD11b after administration [13]. In addition, similar memory impairments were observed in mice, such as worsening of delayed place recognition. Thus, a certain part of the behavioral and memory impairment caused by ATCV-1 is associated with a neuroinflammatory response (Figure 1).

Another study also showed the ability of the ATCV-1 virus to activate macrophages in vitro in a cell culture of mouse RAW264.7 macrophages and a primary culture of inflammatory macrophages from C57BL/6 mice. An hour of incubation with the ATCV-1 virus increased the expression of IL-6 and inducible NO synthase genes, as well as the release of pro-inflammatory cytokines IL-6 and NO. An increase in lactate levels was observed under the influence of the virus, probably due to a shift in metabolism from oxidative phosphorylation towards glycolysis. Such metabolic changes characterize the transition of macrophages from a resting state to a pro-inflammatory M1 phenotype. A characteristic response of mammalian cells to viral infection is the production of interferons and interferon response factors such as IRF7. The expression of the IFN-β and IRF7 genes increased in macrophage cultures after incubation with the ATCV-1 virus, indicating the activation of innate antiviral immunity pathways [14].

To make sure that the virus is not only attached to the cell surface, but is also internalized, ATCV-1 stained with Sytox orange dye was added to the RAW264.7 cell culture (according to D.D. Dunigan, this dye does not affect the ability of the ATCV-1 virus to infect algal cells *Chlorella heliozoae*), and 24 hours later, the cell membrane was stained with CellMask dye to detect intracellular viruses. Confocal microscopy revealed stained viruses inside the cell and outside the cell membrane.

In addition, in a parallel experiment, the number of viruses bound to cells was measured after 1 hour of incubation and removal of the supernatant with unbound viruses. The initial dose was 1 × 10^7^ ATCV-1 PFU (plaque-forming units, measured in a virological plaque formation test on lawns of *Chlorella heliozoae* cells). On average, for RAW264.7, this amount was 0.8 × 10^6^ PFU/culture (8% of the initial amount), and for the primary macrophage culture, 2.7 × 10^5^ PFU/culture (2.7% of the initial amount). However, after 24 hours, the amount of ATCV-1 PFU (isolated from the supernatant and macrophage lysates) was already 2.6 × 10^6^ PFU/culture for RAW264.7 and 3.9 × 10^5^ PFU/culture for the primary macrophage culture. After 72 hours, the amount of PFU was 19 × 10^5^ PFU/culture for RAW264.7 and 5.0 × 10^5^ PFU/culture for primary macrophage culture. Compared to the number of PFUs at the 1-hour time point, a significantly higher number of ATCV-1 PFUs were observed at the 24- and 72-hour time points, indicating the ability of ATCV-1 to persist and slowly replicate in macrophages.

To confirm the replication of the ATCV-1 virus in macrophages on a RAW264.7 culture, the qPCR method measured the expression of mRNA of the viral gene *z280l* encoding the main protein of the ATCV-1 capsid. The average amount of mRNA was 40 copies at the 24-hour time point, 139 copies at the 48-hour time point, and 100 copies at the 72-hour time point. However, to confirm viral replication, it was still necessary to detect the production of ATCV-1 proteins in the RAW264.7 cells. To do this, a rabbit antiserum against the purified ATCV-1 virus was obtained, and when conducting a Western blot with proteins from lysates of RAW264.7 macrophages incubated with ATCV-1, a 55 kDa protein was detected (mass similar to the main protein of the ATCV-1 capsid), the level of which remained constant from 16 h to 66 h after incubation with the virus, and another 17 kDa protein, determined at the point at 16 h and increasing in number from 48 h to 66 h. To determine whether these proteins are synthesized in macrophages or remain in the cells after phagocytosis of the initial amount of virus, a heat-inactivated ATCV-1 virus was used as a control. Despite the fact that at the initial time point 24 h after the addition of the virus, the intensity of the 55 kDa protein bands were equivalent in cultures infected with both the intact and inactivated virus, after 48 h and 72 h, the level of this protein in the cultures with the inactivated virus decreased. In addition, in the case of the inactivated virus, the 17 kDa protein was not detected. Thus, it can be argued that RAW264.7 macrophages, after infection with the intact ATCV-1 virus, do indeed synthesize some viral proteins. However, it is impossible to say for sure whether the synthesis of other proteins of the virus occurs in the cells, and they are not detected because their number is below the level of detection in this experiment, or the antiserum used in the experiment is not able to recognize them.

In addition to the above actions on the cells of macrophage cultures, the ATCV-1 virus has a cytopathic effect on RAW264.7 cells: 24 hours after the addition of the virus, plasma membrane blebbing, karyorrhexis, and apoptosis are observed in some cells (the heat-inactivated ATCV-1 virus did not cause such phenomena). However, the primary culture of inflammatory macrophages did not show a significant level of apoptosis even after 72 h, but the formation of cell membrane outgrowths characteristic of macrophage activation was observed. The RAW264.7 cells, after exposure to the virus, showed an increase in the level of activated caspase 3 and annexin V associated with cell membrane phosphatidylserine, which indicates the activation of programmed cell death through apoptosis, which is probably a protective antiviral mechanism that limits viral replication. However, it is possible that the phagocytosis of apoptotic macrophages by other macrophages promotes the spread of ATCV-1 to them and its persistence in the macrophage population. It was shown that the triggering of apoptosis mechanisms upon the infection of RAW264.7 macrophages with the ATCV-1 virus depends on the activation of the mitogen-activated protein kinases/extracellular signal-regulated kinases (MAPK/ERK) pathway. In Western blot with antibodies to activate phospho-ERK, its increase is observed as soon as 30 minutes after the introduction of the virus; it is detected after 60 minutes, but is absent after 3 hours. The pretreatment of cells with ERK kinase activation inhibitor U0126 prevents RAW264.7 apoptosis caused by the ATCV-1 virus [14].

In this article, a BHK-21 fibroblast culture was also tested for the possibility of ATCV-1 infection, and the results were negative. The question of the ability of the ATCV-1 virus to infect various types of cells remains open: is the virus capable of penetrating the human CNS, infecting neurons or glial cells? Given the data obtained on macrophage cultures, is it possible for ATCV-1 to infect the microglia of the human nervous system, which act as resident macrophages of the CNS and have a common origin with macrophages of other body tissues? As shown, ATCV-1 is capable of persistence and replication in macrophages, which is accompanied by their activation with a pro-inflammatory phenotype, the release of various inflammatory factors, such as IL-6 and NO. In addition, activated macrophages are able to migrate, thus spreading the virus that persists in them. It is known that activated microglia with a pro-inflammatory phenotype can cause serious disturbances in the functioning of CNS cells: it can destroy synaptic connections, cause disturbances in the formation of new synapses, neurogenesis, synaptic plasticity, and cause disturbances in the normal energy and plastic metabolism of neurons and astrocytes, which can trigger the processes of programmed cell death. The activation of microglia and an increase in the level of pro-inflammatory cytokines and NO in the CNS characterizes, among other things, many neurodegenerative diseases.

A recent article [15] elucidates a plausible mechanism of ATCV-1 influence on the development of neurodegenerative diseases, such as Amyotrophic Lateral Sclerosis (ALS). In this article, serum from patients with the sporadic form of ALS was evaluated for anti-ATCV-1 IgG antibodies. The results of subclass-specific anti-ATCV-1 IgG detection showed that sporadic ALS patients (*n* = 18) had significantly higher levels of anti-ATCV-1 IgG1(*p* = 0.0182), but not other IgG subclasses, compared to the healthy control group (*n* = 13).

Further, the authors of this article decided to determine whether ATCV-1 influences the development of motor neuron diseases (MNDs) using a transgenic mouse model of familial ALS, with mice expressing human genetically polymorphic Superoxide Dismutase 1 G93A. SOD1-G93A-transgenic mice or C57Bl/6 control mice were intracranially injected at day 35 of age with ATCV-1, heat-killed ATCV-1, PBS, or polyinosinic:polycytidylic acid (polyI:C), a stimulant of the antiviral immune response. The SOD1-G93A mice, injected with active ATCV-1, showed more rapid motor deterioration compared to the saline-treated transgenic mice. However, active ATCV-1 infection did not significantly affect mortality in this group. Inactivated ATCV-1 did not affect motor loss or survival, so only active ATCV-1 hastened the onset of MND in SOD1-G93A mice. An interesting finding in this study was that polyI:C injection significantly lengthened survival time, but had a minimal effect on motor loss onset. 

It is known that the ATCV-1 genome encodes a functional SOD-1 protein, which converges with findings that ATCV-1 increases motor deterioration in SOD1-G93A mice. To study the effects of ATCV-1 SOD1 on macrophage expression of immune factors, such as IL-6, IL-10, NO, and Interferon-Stimulated Genes (ISGs) promotor activity, RAW264.7 Lucia cells were transfected with a plasmid vector that expresses ATCV-1 SOD1. The results of assays with transfected cells indicate that ATCV-1 SOD1 augments the production of IL-6, IL-10, NO, and ISG expression, which are associated with the development of ALS in humans and MND in SODG93A transgenic mice.

## 4. Features of Finding ATCV-1 in the Environment and Possible Routes of Transmission

Chloroviruses, to which ATCV-1 belongs, are widely distributed in nature. They are found in freshwater bodies throughout the world, and have been isolated from freshwater sources in Europe, Asia, Australia, and North and South America. The titer of chloroviruses in freshwater samples depends on the time of year and the location of the reservoir, but usually ranges from 1 to 100 PFU / ml; however, much higher values have been recorded, up to 100,000 PFU/ml. Chloroviruses infect chlorella-like algae (*Chlorella* sp. and *Micractinium* sp.), with a specific virus species infecting a specific algal species. The host for ATCV-1 is *Chlorella heliozoae*. Algae are able to establish endosymbiotic relationships with larger protists. Thus, the host for *Chlorella heliozoae* is *Acanthocystis turfacea*. One protist host can contain up to several hundred unicellular algae. *Chlorella* sp. in an endosymbiotic state is resistant to chlorovirus infection, whereas symbiont-free chlorella are susceptible to infection [16]. The titers of ATCV-1 in a body of water can vary greatly throughout the year depending on the temperature, ultraviolet (UV) radiation intensity, and the abundance of *Chlorella heliozoae*. The consumption of protists hosting algae by small crustaceans can indirectly increase the titer of chloroviruses. In this case, a large number of single-celled green algae are released, which become vulnerable to viral infection due to the disruption of endosymbiosis. However, an increase in chlorovirus titers in the presence of predatory small crustaceans was only shown for viruses of *Chlorella variabilis* strains that are symbionts of *Paramecium bursaria* [17]. Since the host of the ATCV-1 virus is *Chlorella heliozoae* and it is a symbiont of the *Heliozoae* species, it cannot be asserted that the concentration of *Chlorella heliozoae* viruses increases in the presence of predators. However, the study of this issue is extremely important in the context of ATCV-1 as a possible infectious agent for human health.

A distinctive feature of ATCV-1 is one of the lowest decay rates of viral particles among all algal viruses. The highest decay rate is observed in the summer period of the year, which is characterized by a high intensity of UV-B radiation, high daytime temperatures, increased activity of extracellular bacterial nucleases and other enzymes, and the consumption of viruses by nanoflagellates. However, ATCV-1 and some other chloroviruses are able to repair DNA damage from exposure to UV radiation using virus-encoded UV-dependent DNA glycosylase–pyrimidine lyase, which is able to eliminate thymidine dimers as well as other viral repair enzymes. The lowest rate of virus decay is observed in winter. Moreover, it has been shown that ATCV-1 can tolerate complete freezing for extended periods of time, allowing it to survive in season-related freezing waters [2,18].

Although the route of ATCV-1 transmission through contact with water containing infected algae *Chlorella heliozoae* appears obvious at first glance—swimming in a reservoir and swallowing water through the choanae into the nasal cavity, as well as passage of water through the nostrils—there are no reliable data on transmission routes. Other possible modes of human infection are not ruled out. When consuming water contaminated with infected *Chlorella heliozoae*, it can assume the alimentary route of ATCV-1 infection. Such water with ATCV-1 can enter the nasal cavity through the choanae, and then the virus can enter the brain through the olfactory nerves. Infection via Chlorella-based nutritional supplements is another possible alimentary route of ATCV-1 infection. Chlorella is commercially produced and distributed worldwide as a nutritional supplement and contains larger amounts of vitamins, including D, B9, and B12 [19]. For nutritional supplements, certain species of *Chlorella* sp. are used, for example, in the EU, where traditionally there are three species: *Chlorella pyrenoidosa, Chlorella vulgaris*, and *Chlorella luteoviridis*. But as Champenois et al.’s analysis showed, under real production conditions, control is not carried out properly, and *Chlorella* species are incorrectly identified [20]. This means that potentially chlorovirus-infected *Chlorella* can also become added into nutritional supplements, which is a potential threat of ATCV-1 infection both in the alimentary route and by the accidental inhalation of a dry suspension of *Chlorella* from a nutritional supplement when opening a food product package.

## 5. Conclusions

The discovery of the ATCV-1 virus as part of the human oropharyngeal virome prompts a new look at viruses from freshwater and other natural sources. There have been no previous studies describing the similar effect of viruses that infect algae on the human body, except one study that describes the isolation of viruses of the *Phycodnaviridae* family in the cervico-vaginal secretion of women and their association with gynecological diseases [21]. Topical microbicides, which have already shown their effectiveness and preventive significance in stages II and III of clinical trials for HIV prevention, can protect the mucous membranes, the point of entry for a viral infection, from new unknown viral threats [22]. The scientific community should pay attention to reviewing the problems of the interaction of mucosal immunity with the virome of human cavities, including the nasal cavity, and its possible correction with the help of immunoactive microbicides. As shown above, ATCV-1 transmission routes also remain unexplored. Further research will help to identify the routes of infection, the cell types in which ATCV-1 can persist, and the pathological mechanisms of the virus’s effect on the human body.

## Figures and Tables

**Figure 1 tropicalmed-08-00040-f001:**
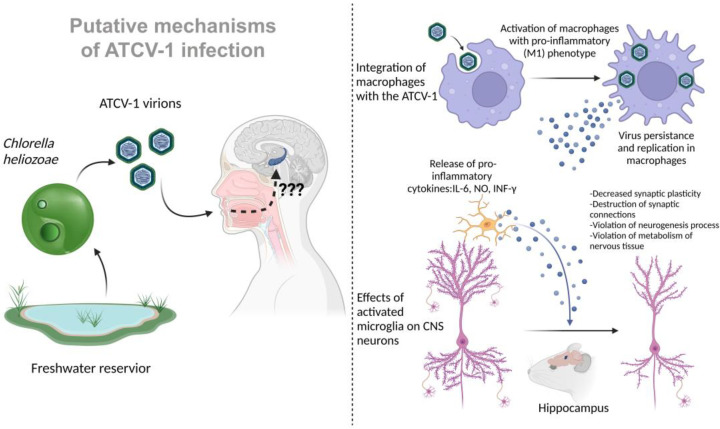
Putative mechanism of damage to the human central nervous system during infection with ATCV-1. (**Left**) A suggested route of human infection with ATCV-1 is shown by swimming in a freshwater reservoir containing infected *Chlorella heliozoae*. There is currently no precise information on how ATCV-1 enters and damages the CNS (indicated by a question mark “???”), such as by crossing the blood–brain barrier or through an indirect effect linked to systemic inflammation in the CNS. (**Right**) The putative mechanisms of ATCV-1 nerve tissue damage, which have been mouse-modeled in vivo, are depicted. IL-6—interleukin-6; NO—nitric oxide; IFN-γ—interferon gamma.

## Data Availability

Not applicable.

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
