# Peer review of "The Role of Chloroviruses as Possible Infectious Agents for Human Health: Putative Mechanisms of ATCV-1 Infection and Potential Routes of Transmission"

_tropicalmed, 2023, doi:10.3390/tropicalmed8010040_

Round 1

Reviewer 1 Report

The authors did a satisfactory job in their review of a scant number of publications related to association of chlorovirus ATCV-1 and human (animal) psychiatric and mental disorders. However, the value of this work was greatly compromised by a lack of accuracy and clarity in the author’s presentation that made reading and reviewing the manuscript quite difficult and decreases the credibility of their work. The manuscript needs extensive English editing. In a few instances it was impossible to distinguish between the authors opinion/conclusion and publications the authors used in their review.

 It seems that the review title “The role of chloroviruses as new dangerous infectious agents for human health: mechanisms of ATCV-1 infection and potential routes of transmission” is exaggerated quite a bit and as of currently available data it misrepresents chlorovirus role in human mental and psychiatric disorders. The phrase “dangerous infectious agents’” needs to be removed from the title and throughout the manuscript. At this point one can talk only about association of ATCV-1 virus with mental and psychiatric disorders. 

Some examples of areas needing revision: 

L24-25: “Is the fact that one can detect the DNA of the virus associated with swimming in bodies of water with a large amount of green algae or drinking water from such sources without sufficient purification?”

It is not clear what the authors wanted to say. 

L. 30-31: “…algal viruses as potential infectious agents and a significant risk factor for human nervous system diseases.”

It is too speculative. 

L 38-39: “… with a particular algae species infecting a specific virus species.”

It is the other way around: the viruses infect algae species. 

L. 138-139:  The figure 1 title needs to be changed to “PUTATIVE (or PRESUMPTIVE) mechanism of damage to the human central nervous system during infection with the ATCV-1. IL-6 – interleukin-6; NO – nitric oxide; IFN-γ – interferon gamma. 

L.162: “…plaque formation test on a monolayer of Chlorella heliozoae cells”.

Chlorovirus plaque assays are done on lawns of chlorella cells and they are not a monolayer.  

L. 239-240: “Algae hosts of chloroviruses, ways to establish endosymbiotic relationships with larger protists, for Chlorella heliozoae, the host is a representative of the centrohelid sunflower Acanthocystis turfacea.“

Don’t understand what the authors wanted to say.

L. 242-243: “Algae Chlorella sp. in a state of endosymbiosis with protists, are resistant to infection by chloroviruses, while free-swimming representatives are susceptible to infection.”

Chlorellas are not free-swimming alga. It is better to say “symbiont free chlorella”.

L. 244-248: “The titers of ATCV-1 in a water body can vary greatly throughout the year depending on temperature, ultraviolet radiation intensity, abundance of Chlorella heliozoae, and representation of other microorganisms. For example, the consumption of protists hosting algae by small crustaceans can indirectly increase the titer of chloroviruses [15].”

An increase of chlorovirus titers in presence of predators was shown only for viruses of Chlorella variabilis strains that are symbionts of Paramecium bursaria.  The host of ATCV-1 virus is Chlorella heliozoae and it is symbiont of Heliozoae species, that makes it too speculative to claim that concentration of Chlorella heliozoae viruses goes up in the presence of predators. 

Author Response

Reviewer 1

The authors did a satisfactory job in their review of a scant number of publications related to association of chlorovirus ATCV-1 and human (animal) psychiatric and mental disorders. However, the value of this work was greatly compromised by a lack of accuracy and clarity in the author’s presentation that made reading and reviewing the manuscript quite difficult and decreases the credibility of their work. The manuscript needs extensive English editing. In a few instances it was impossible to distinguish between the authors opinion/conclusion and publications the authors used in their review. It seems that the review title “The role of chloroviruses as new dangerous infectious agents for human health: mechanisms of ATCV-1 infection and potential routes of transmission” is exaggerated quite a bit and as of currently available data it misrepresents chlorovirus role in human mental and psychiatric disorders. The phrase “dangerous infectious agents’” needs to be removed from the title and throughout the manuscript. At this point one can talk only about association of ATCV-1 virus with mental and psychiatric disorders. 

Dear Reviewer 1,

Thank you for reviewing our manuscript! We really appreciate your comments and advice! We have corrected the manuscript and proofread the English language.

We have accepted your comment and changed the title. We also changed "dangerous infectious agents " to "possible infectious agents " from the text of the manuscript.

We also proofread the text with an English-speaking reviewer to improve the perception of the text by readers.

Some examples of areas needing revision: 

L24-25: “Is the fact that one can detect the DNA of the virus associated with swimming in bodies of water with a large amount of green algae or drinking water from such sources without sufficient purification?”

It is not clear what the authors wanted to say. 

The sentence has been removed.

L. 30-31: “…algal viruses as potential infectious agents and a significant risk factor for human nervous system diseases.”

It is too speculative. 

We agree; the sentence has been removed.

L 38-39: “… with a particular algae species infecting a specific virus species.”

It is the other way around: the viruses infect algae species. 

Thanks for your comment! The typo has been corrected.

L. 138-139:  The figure 1 title needs to be changed to “PUTATIVE (or PRESUMPTIVE) mechanism of damage to the human central nervous system during infection with the ATCV-1. IL-6 – interleukin-6; NO – nitric oxide; IFN-γ – interferon gamma. 

Agree with comment. Changes in the figure 1 title have been made.

L.162: “…plaque formation test on a monolayer of Chlorella heliozoae cells”.

Chlorovirus plaque assays are done on lawns of chlorella cells and they are not a monolayer.

Thanks for your comment! The typo has been corrected.

L. 239-240: “Algae hosts of chloroviruses, ways to establish endosymbiotic relationships with larger protists, for Chlorella heliozoae, the host is a representative of the centrohelid sunflower Acanthocystis turfacea.“

Don’t understand what the authors wanted to say.

We have changed the incorrectly formulated sentence. Now it looks like this: ‘Algae are able to establish endosymbiotic relationships with larger protists. Thus, the host for Chlorella heliozoae is a representative of the centrohelid Acanthocystis turfacea.’

We also proofread the text with an English-speaking reviewer to improve the perception of the text by readers.

L. 242-243: “Algae Chlorella sp. in a state of endosymbiosis with protists, are resistant to infection by chloroviruses, while free-swimming representatives are susceptible to infection.”

Chlorellas are not free-swimming alga. It is better to say “symbiont free chlorella”.

Agree with comment. Changes made; citation added. 

L. 244-248: “The titers of ATCV-1 in a water body can vary greatly throughout the year depending on temperature, ultraviolet radiation intensity, abundance of Chlorella heliozoae, and representation of other microorganisms. For example, the consumption of protists hosting algae by small crustaceans can indirectly increase the titer of chloroviruses [15].”

An increase of chlorovirus titers in presence of predators was shown only for viruses of Chlorella variabilis strains that are symbionts of Paramecium bursaria.  The host of ATCV-1 virus is Chlorella heliozoae and it is symbiont of Heliozoae species, that makes it too speculative to claim that concentration of Chlorella heliozoae viruses goes up in the presence of predators.

Reviewer agreement. The paragraph has been rewritten: “Since the host of ATCV-1 virus is Chlorella heliozoae and it is a symbiont of Heliozoae species, it cannot be asserted that the concentration of Chlorella he-liozoae viruses increases in the presence of predators. However, the study of this issue is extremely important in the context of ATCV-1 as a possible infectious agent for human health.” Please see lines 3200-356.

Reviewer 2 Report

The authors, who to my knowledge have never worked on the chloroviruses, provide an opinion manuscript on the role that certain chloroviruses play in motor neuron diseases. The authors refer to the virus ATCV-1 as a "new dangerous infectious agent" of humans. This reviewer thinks that it is too early to say that ATCV-1 is significantly involved in these motor neuron diseases and recommends removing the word dangerous. Change "new dangerous infectious agents" to "possible infectious agents for human health" in the title. Likewise, remove "dangerous" through out the manuscript. For example, just in the abstract - line 22, change "dangerous" to "possible", line 27, remove "dangerous", line 31, change "significant" to "possible". Current research has reported that some chlorovirus ATCV-1-like genome sequences have been found in humans with certain neurological issues. In addition, there is a recent report that exposure of SOD1-G93A [a mutation found in some ALS patients] transgenic mice to ATCV-1 accelerates motor deterioration. This recent publication [Petro et al., (2022) Frontiers in Neurology 13, 821166] was not cited in the current opinion article. 

The manuscript requires a good editor to look it over. For example in line 38, the authors state "....., with a particular algae species infecting a specific virus species." [ see also line 238]. This is backwards, the viruses infect the algae. 

In line 41, the name of ATCV-1 is in italics - ICTV says virus names should not be in italics. Also see line 49.

Some specific suggestions.

line 167. After 72 hours, the amount of PFU was 19 x105, not 1.9 x 105.

line 171. add "slowly" replicate in macrophages. The virus titer only goes from 1.7 x 105 to 5.0 x 105 in 72 hours.

line 238. One group of chloroviruses infect algae that are in the genus Micractinium. I would say chlorella-like algae.

line 239-240. Not sure what the authors are trying to say.

line 241. remove "sunflower"?

line 243. Chlorella-like algae are not "free-swimming". They lack flagella and cilia.

line 255. add "...using 'virus-encoded' UV-dependent..."

Author Response

Reviewer 2

The authors, who to my knowledge have never worked on the chloroviruses, provide an opinion manuscript on the role that certain chloroviruses play in motor neuron diseases. The authors refer to the virus ATCV-1 as a "new dangerous infectious agent" of humans. This reviewer thinks that it is too early to say that ATCV-1 is significantly involved in these motor neuron diseases and recommends removing the word dangerous. Change "new dangerous infectious agents" to "possible infectious agents for human health" in the title. Likewise, remove "dangerous" through out the manuscript. For example, just in the abstract - line 22, change "dangerous" to "possible", line 27, remove "dangerous", line 31, change "significant" to "possible".

Dear Reviewer 2,

Thank you for reviewing our manuscript! We really appreciate your comments and advice! We have corrected the manuscript and proofread the English language.

Although we have not really worked with chloroviruses, our team—the Department of General Hygiene at Sechenov University—are representatives of WHO experts on drinking water safety. In this manuscript, we want to identify a potential problem caused by chlorovirus contamination of water and food products. I hope future readers will find this ‘Opinion’ interesting!

We have accepted your comment and changed the title and changed "dangerous/ significant " to "possible" from the text of the manuscript.

Current research has reported that some chlorovirus ATCV-1-like genome sequences have been found in humans with certain neurological issues. In addition, there is a recent report that exposure of SOD1-G93A [a mutation found in some ALS patients] transgenic mice to ATCV-1 accelerates motor deterioration. This recent publication [Petro et al., (2022) Frontiers in Neurology 13, 821166] was not cited in the current opinion article. 

Thanks for your important comment. We got acquainted with the specified article; it was published even before the writing of our manuscript. We have added additional information to our manuscript. Pls, see lines 280-306

The manuscript requires a good editor to look it over. For example in line 38, the authors state "....., with a particular algae species infecting a specific virus species." [ see also line 238]. This is backwards, the viruses infect the algae. 

Thanks for your comment! The typo has been corrected.

In line 41, the name of ATCV-1 is in italics - ICTV says virus names should not be in italics. Also see line 49.

Changes applied.

Some specific suggestions.

line 167. After 72 hours, the amount of PFU was 19 x105, not 1.9 x 105.

Thank you! Changes have been made.

line 171. add "slowly" replicate in macrophages. The virus titer only goes from 1.7 x 105 to 5.0 x 105 in 72 hours.

Changes have been made.

line 238. One group of chloroviruses infect algae that are in the genus Micractinium. I would say chlorella-like algae.

Thanks for your comment, changes have been made.

line 239-240. Not sure what the authors are trying to say.

We have changed the incorrectly formulated sentence. Now it looks like this: ‘Algae are able to establish endosymbiotic relationships with larger protists. Thus, the host for Chlorella heliozoae is a representative of the centrohelid Acanthocystis turfacea.’

We also proofread the text with an English-speaking reviewer to improve the perception of the text by readers.

line 241. remove "sunflower"?

Changes applied.

line 243. Chlorella-like algae are not "free-swimming". They lack flagella and cilia.

Agree with comment. Changes made; citation added. 

line 255. add "...using 'virus-encoded' UV-dependent..."

Thank you! Changes applied.
